# Hypolipogenic Effect of Shikimic Acid Via Inhibition of MID1IP1 and Phosphorylation of AMPK/ACC

**DOI:** 10.3390/ijms20030582

**Published:** 2019-01-29

**Authors:** Moon Joon Kim, Deok Yong Sim, Hye Min Lee, Hyo-Jung Lee, Sung-Hoon Kim

**Affiliations:** College of Korean Medicine, Kyung Hee University, Seoul 02447, Korea; pigcross@naver.com (M.J.K.); simdy0821@naver.com (D.Y.S.); glansy555@gmail.com (H.M.L.); hyonice77@naver.com (H.-J.L.)

**Keywords:** shikimic acid, MID1IP1, AMPK, hepatocellular carcinoma (HCC), 3T3-L1, lipogenesis

## Abstract

Although shikimic acid from *Illicium verum* has antioxidant, antibacterial, anti-inflammatory, and analgesic effects, the effect of shikimic acid on lipogenesis has not yet been explored. Thus, in the present study, hypolipogenic mechanism of shikimic acid was examined in HepG2, Huh7 and 3T3-L1 adipocyte cells. Shikimic acid showed weak cytotoxicity in HepG2, Huh7 and 3T3-L1 cells, but suppressed lipid accumulation in HepG2, Huh7 and 3T3-L1 cells by Oil Red O staining. Also, shikimic acid attenuated the mRNA expression of de novo lipogenesis related genes such as FAS, SREBP-1c, and LXR-α in HepG2 cells by RT-PCR analysis and suppressed the protein expression of SREBP-1c and LXR-α in HepG2 and 3T3-L1 cells. It should be noted that shikimic acid activated phosphorylation of AMP-activated protein kinase (AMPK)/Aacetyl-coenzyme A carboxylase (ACC) and reduced the expression of MID1 Interacting Protein 1 (MID1IP1) in HepG2, Huh7 and 3T3-L1 cells. Conversely, depletion of MID1IP1 activated phosphorylation of AMPK, while overexpression of MID1IP1 suppressed phosphorylation of AMPK in HepG2 cells. However, AMPK inhibitor compound c did not affect the expression of MID1IP1, indicating MID1IP1 as an upstream of AMPK. Taken together, our findings suggest that shikimic acid has hypolipogenic effect in HepG2 and 3T3-L1 cells via phosphorylation of AMPK/ACC and inhibition of MID1IP1 as a potent candidate for prevention or treatment of fatty liver and hyperlipidemia.

## 1. Introduction

Fatty liver disease is caused by excessive fat accumulation, leading to progressive liver fibrosis and cirrhosis with features of metabolic syndrome including insulin resistance [1,2]. Excessive intake of alcohol or fatty food can induce alcoholic or nonalcoholic fatty liver diseases (NAFLD) [3] by promoting de novo fatty acid synthesis through downregulation of AMP-activated protein kinase (AMPK), an important hepatic transcriptional regulator, and then its downstream acetyl CoA carboxylase (ACC) [4,5].

It is well documented that De Novo lipogenesis (DNL) can induce hepatic steatosis and/or hypertriglyceridemia, and also cause steatohepatitis by saturated fatty acids including palmitate [6]. Also, DNL is regulated mainly by two key transcription factors such as sterol regulatory element-binding protein 1c (SREBP1c), which is activated by insulin and liver X receptor α (LXR-α), and carbohydrate regulatory element-binding protein (ChREBP) [7,8].

The MID1 Interacting Protein 1 (MID1IP1) also known as MIG12 or S14 has been implicated in lipogenesis in mammals [9]. Hence, Kim et al. [10] reported that MID1IP1 regulates and binds to acetyl-CoA carboxylase (ACC), the first committed enzyme in fatty acid (FA) synthesis, and induces ACC polymerization during increased haptic FA synthesis. Furthermore, Inoue et al. [11] claimed that MID1IP1 regulates LXR ligand and glucose, resulting in triglyceride accumulation and fatty liver.

Since shikimic acid is a natural compound isolated from the Japanese plant, *Illicium verum* [12], and seeds of *Liquidambar styraciflua* (sweetgum) abundant in North America [13] and Chinese star anise (*Illicium verum*), shikimic acid has been used as a base material for production of oseltamivir (Tamiflu) [14]. Also, though shikimic acid is known to have anti-diabetic [15], antibacterial [16], anti-inflammatory [17], analgesic [18], antioxidant [18], and anti-thrombogenic [19] effects, its hypolipogenic mechanism has never been reported. Thus, in the present study, hypolipogenic mechanism of shikimic acid was elucidated in HepG2 and Huh7 hepatocellular carcinoma HCC cells and 3T3-L1 adipocytes in association with AMPK/ACC and MID1IP1 signaling axis.

## 2. Results

### 2.1. Shikimic Acid Exerted Weak Cytotoxicity in HepG2 and Huh7 Cells and 3T3-L1 Cells

The cytotoxicity of shikimic acid (Figure 1b,c) was evaluated in HepG2 and Huh7 cells and 3T3-L1 cells by MTT assay after the cells were treated with various concentrations of shikimic acid (0, 10, 20, 40, 80, 160 μM). As shown in Figure 1b,c, the viability of HepG2 and Huh7 or 3T3-L1 cells was maintained up to 70% of untreated control even at the concentration of 160 μM. 

### 2.2. Shikimic Acid Reduced the Number of Lipid Droplets in HCCs

To confirm the hypolipidemic effect of shikimic acid, Oil Red O staining was conducted in shikimic acid-treated HCC cells. As shown in Figure 2a, lipid droplets were significantly decreased in a concentration-dependent manner in HepG2 and Huh7 cells by shikimic acid. Similarly, shikimic acid reduced lipid accumulation in 3T3-L1 adipocytes as well (Figure 2b).

### 2.3. MID1IP1 Depletion Suppressed Proliferation and the Expression of SREBP-1c and FAS in HepG2 Cells

MID1IP1 was highly expressed in HepG2 cells better than 3T3-L1 and other cancer cell lines (Figure 3a). To assess the effect of MID1IP1 on lipogenesis-related genes, RT-qPCR analysis was conducted in HepG2 cells. As shown in Figure 3b, mRNA expression of MID1IP1 was attenuated to one quarter of untreated control in HepG2 cells transfected with siRNA plasmid (Figure 3b). Interestingly, the proliferation was weakly reduced in HepG2 cells compared to untreated control by MID1IP1 siRNA transfection (Figure 3c), whereas depletion of MID1IP1 by siRNA transfection method attenuated the expression of SREBP-1c and FAS in HepG2 cells (Figure 3d,e).

### 2.4. Shikimic Acid Downregulated MID1IP1 Expression Level by Phosphorylation of AMPK in HCCs and Adipocytes

To further examine the hypolipogenic effect of shikimic acid, western blot was conducted to estimate the expression level of lipogenesis-related proteins such as p-AMPKα, AMPKα, p-ACC, ACC, MID1IP1, LXR-α and SREBP-1c in HepG2 cells, Huh7 cells and 3T3-L1 adipocytes after shikimic acid treatment for 24 h. Shikimic acid reduced the expression level of MID1P1, LXR-α and SREBP-1c. However, shikimic acid significantly upregulated phosphorylation of AMPKα and ACC in HepG2 cells and adipocytes (Figure 4a,b).

### 2.5. Pivotal Role of MID1IP1 in Shikimic Acid Regulated Lipogenesis in HepG2 and AML-12 Cells

To examine the role of MID1IP1 in shikimic acid-regulated lipogenesis-related genes, overexpression or depletion plasmid of MID1IP1 and AMPK inhibitor compound c were used in AML-12 and HepG2 cells. As shown in Figure 5a, overexpression of MID1IP1-reduced phosphorylation of AMPK by shikimic acid (80 μM) in AML-12 cells (Figure 5b), whereas depletion of MID1IP1 activated phosphorylation of AMPK/ACC in HepG2 cells (Figure 5c). However, AMPK inhibitor compound c did not affect expression of MID1IP1 in HepG2 cells (Figure 5d).

## 3. Discussion

Herein, hypolipogenic mechanism of shikimic acid from *I. verum* was examined in HCC cells and 3T3-L1 adipocyte cells. It is well known that products by hepatic de novo lipogenesis, esterification of plasma free fatty acids or increased dietary fat intake are critically involved in development of NAFLD [3,20]. Shikimic acid showed weak cytotoxicity in HCC cells and 3T3-L1 cells. To confirm hypolipogenic effect of shikimic acid, Oil red O staining assay that has been applied for evaluation of liver steatosis and lipid metabolism [21] was conducted in lipogenic HepG2, weak lipogenic Huh7 and adipogenic 3T3-L1 cells. Shikimic acid suppressed lipid accumulation in HepG2 and 3T3-L1 cells, implying hypolipogenic potential of shikimic acid. 

Accumulating evidences reveal that SREBP-1c primarily regulates FAS, whereas liver X receptors (LXR) regulate transcription of SREBP-1c through LXR response element (LXRE) for cholesterol homeostasis and lipogenesis [22,23,24]. Consistently, RT-qPCR analysis showed that shikimic acid attenuated the mRNA expression of de novo lipogenesis-related genes such as FAS, SREBP-1c, and LXR-α in HepG2 cells. Likewise, shikimic acid attenuated the protein expression of SREBP-1c and LXR-α in HepG2 and 3T3-L1 cells, indicating shikimic acid inhibits lipogenesis-related genes both at mRNA and protein levels. 

Emerging evidence shows that AMPK, a sensor of cellular energy charge and a “metabolic master switch”, enhances fatty acid oxidation by lowering the concentration of malonyl coenzyme A (malonyl CoA) and also modulates the concentration of malonyl CoA by concurrently phosphorylating and inhibiting acetyl CoA carboxylase (ACC) alpha or beta [25,26]. Notably, shikimic acid activated phosphorylation of AMPK and its downstream ACC in HepG2, Huh7 and 3T3-L1 cells, demonstrating the critical role of pAMPK/pACC. 

Recent studies reveal that MID1IP1, known as MIG12 or S14, activates ACC for fatty acid synthesis and also controls triglyceride accumulation in fatty liver [9,10,27]. Interestingly, shikimic acid attenuated expression of MID1IP1 in HepG2 cells at mRNA and protein levels, implying antiadipogenic potential of shikimic acid. Conversely, knockdown of MID1IP1 activated phosphorylation of AMPK, whereas overexpression of MID1IP1 reduced phosphorylation of AMPK in HepG2 cells. In contrast, AMPK inhibitor compound c did not affect the expression of MID1IP1, indicating that MID1IP1 can be an upstream of AMPK. Nonetheless, it is still necessary to perform further experiments for detailed mechanism using IP, genome editing by way of CRISRP/Caspase9 assay, RNA editing methods and in animal study for future clinical trials.

Overall, our findings provide evidence that shikimic acid has a hypolipogenic effect in HepG2 and 3T3-L1 cells via phosphorylation of AMPK/ACC and inhibition of MID1IP1 as a potential candidate for prevention or treatment of fatty liver and hyperlipidemia (Figure 6).

## 4. Materials and Methods

### 4.1. Reagents, Antibodies and Plasmids

Shikimic acid, Oil-red-O powder, SREBP-1c(SREBF1), LXR-α and β-actin were purchased from Sigma-Aldrich (St. Louis, MO, USA). Shikimic acid was dissolved in distilled water according to manufacturer’s instruction. Lipofectamine 2000 reagent was purchased from Invitrogen (Carlsbad, CA, USA). Roswell Park Memorial Institute (RPMI) 1640, Dulbecco’s modified Eagle’s medium (DMEM), and fetal bovine serum (FBS) were purchased from Welgene (Daegu, Gyeongsangbuk-do, Korea). Antibodies of p-AMPKα, AMPKα, p-ACC and ACC were purchased from Cell Signaling Technology (Beverly, MA, USA). Antibody of MID1IP1 was purchased from Abcam (Abcam, Cambridge, Cambridgeshire, United Kingdom). Primers for MID1IP1, SREBP-1c, LXR-α and FAS were obtained from Bioneer (Bioneer, Daejun, Korea). MID1IP1 siRNA and overexpression plasmids were purchased from Addgene (Addgene, Cambridge, MA, USA). 

### 4.2. Cell Lines and Culture

HepG2 liver hepatocellular cancer (ATCC^®^ HB-8065™) was purchased from the American Type Culture Collection (ATCC, Manassas, VA, USA). Huh7 liver hepatocellular cancer, AML-12 liver normal cells and preadipocyte 3T3-L1 cells were obtained from the Korean Cell Line Bank (KCLB, Seoul, Korea). HepG2 cells, AML-12 and preadipocyte 3T3-L1 cells were cultured in DMEM supplemented with 10% FBS and 1% antibiotics, and Huh7 cells were maintained in RPMI 1640. All cells were incubated at 37 °C under condition of relative humidity and 5% CO_2_.

### 4.3. Cell Viability Assay

The cytotoxicity of shikimic acid was evaluated in HepG2, Huh7 and 3T3-L1 cells by using colorimetric 3-(4,5-dimethylthiazol-2-yl)-2,5-diphenyltetrazolium bromide (MTT) assay (Sigma, St. Louis, MO, USA). Briefly, cells were treated by various concentrations (0, 10, 20, 40, 80, 160 μM) of shikimic acid for 24 h and then were exposed to MTT (1 mg/mL) for 2 h. Then optical density (OD) was measured using a microplate reader (Tecan, Switzerland) at a wavelength of 570 nm. Cell viability was calculated as a percentage of viable cells in a shikimic acid-treated group versus untreated control. 

### 4.4. Adipogenic Differentiation Induction

The preadipocyte 3T3-L1 cells were incubated onto 6-well plates at 0.8 × 10^5^ cells/well in DMEM supplemented with 10% FBS and 1% antibiotics for two days. To induce differentiation, 3T3-L1 preadipocytes were incubated in DMEM with 1 μM dexamethasone, and 1 μg/mL of insulin, and 0.5 mM isobutylmethylxanthine (IBMX) (Sigma-Aldrich, St. Louis, MO, USA) for two days and were replaced by fresh normal medium containing 1 μg/mL of insulin.

### 4.5. Oil-Red-O Staining

The 3T3-L1 cells were treated with or without shikimic acid (40, 80 μM), fixed with 4% paraformaldehyde at room temperature for 30 min and washed with distilled water twice and 60% isopropanol. The cells were stained for 20 min at room temperature by immersion with Oil-Red-O solution (Sigma-Aldrich, St. Louis, MO, USA) and then were washed with distilled water four times. The plate was photographed using a camera connected to an Axio observer a1 inverted microscope (Zeiss, Germany).

### 4.6. Western Blotting

For protein extraction, HCCs or 3T3-L1 cells treated with or without shikimic acid (40, 80 μM) were lysed with RIPA lysis buffer (Thermo) with protease inhibitor. Twenty to thirty micrograms of total protein were separated on SDS-PAGE and electrotransferred to nitrocellulose blotting membranes (Amersham Biosciences, Buckinghamshire, UK). The membranes were blocked with 3% non-fat dry milk in TBST and probed with antibodies of SREBP-1c, LXR-α, p-AMPKα, AMPKα, p-ACC, ACC and β-actin at 4 °C. After washing, the membrane was incubated with horseradish peroxidase (HRP)-conjugated secondary antibodies, and protein expression was examined by enhanced chemiluminescence (ECL) (GE Health Care Biosciences, Piscataway, NJ, USA).

### 4.7. Quantitative Real-Time Polymerase Chain Reaction (qRT-PCR)

The total RNA of cells was isolated from HepG2 cells using QIAZOL lysis reagent (QIAZEN, Venlo, Netherlands). After synthesis process of cDNA by using M-MLV reverse transcriptase (Promega, WI, USA), the mRNA levels were measured by qRT-PCR with the light cycler TM instrument (Roche Applied Sciences, IN, USA) according to manufacturer’s protocol. The mRNA level of GAPDH was used to normalize the expression of genes of interest. Primers of MID1IP1, SREBP-1c and FAS were purchased from Bioneer. The sequences of these primers used are as follows (Table 1): 

Each sample was tested in triplicates, and relative gene expression data were analyzed by means of the 2^−∆CT^ method.

### 4.8. RNA Interference

The AML-12 cells were transfected with MID1IP1 overexpression or siRNA plasmid using X-treme-transfection reagent (Sigma-Aldrich) according to manufacturer’s protocol for next experiment. The mixtures of the MID1IP1 overexpression or siRNA plasmid and X-treme-transfection reagent were incubated for 25 m, and the cells were incubated at 37 °C for 48 h before exposure to 80 μM of shikimic acid for 24 h. 

### 4.9. Statistical Analysis

The data were expressed as means ± standard deviation (SD) of three replications per experiment. Analysis of variance (ANOVA) was conducted to determine the significant differences between two groups. *p* < 0.05 was considered significant.

## 5. Conclusions

Shikimic acid has hypolipogenic effect in HepG2 and 3T3-L1 cells via phosphorylation of AMPK/ACC and inhibition of MID1IP1 as a potent candidate for prevention or treatment of fatty liver and hyperlipidemia.

## Figures and Tables

**Figure 1 ijms-20-00582-f001:**
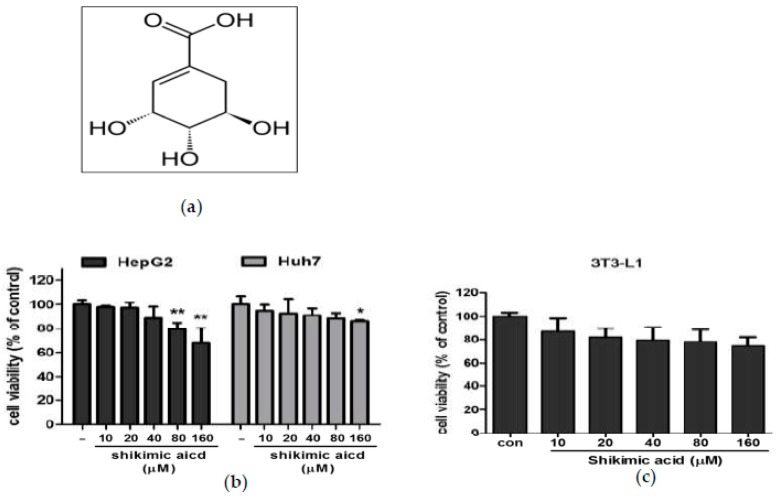
Chemical structure of shikimic acid and its effect on cytotoxicity in HepG2, Huh7 and 3T3 cells. (**a**) Chemical structure of shikimic acid. (**b**) Cytotoxicity of shikimic acid in HepG2 and Huh7 cells. The cells were treated with shikimic acid for 24 h and cytotoxicity was evaluated by 3-(4,5-dimethylthiazol-2-yl)-2,5-diphenyltetrazolium bromide MTT assay (**c**) Cytotoxicity of shikimic acid in 3T3-L1 cells. Results represent means ± S.D. from three independent experiments. * *p* < 0.05, ** *p* < 0.01 versus control.

**Figure 2 ijms-20-00582-f002:**
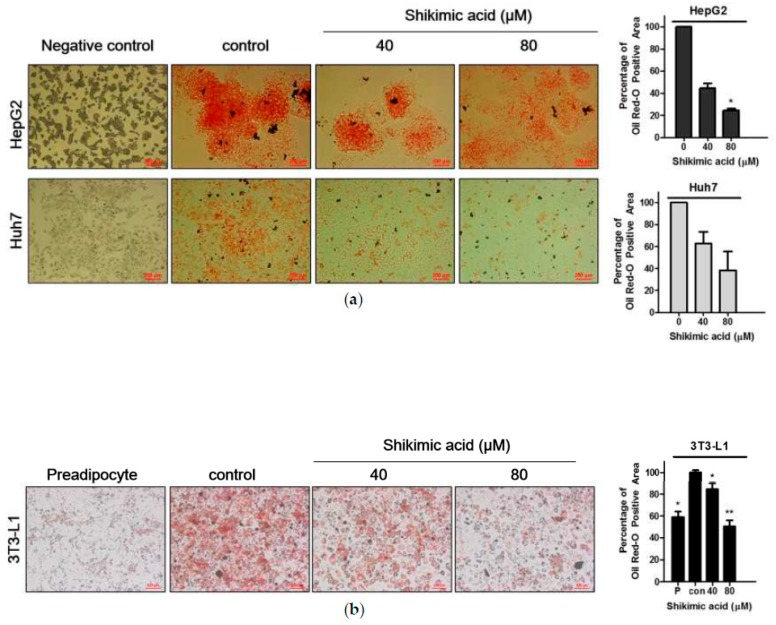
Effect of shikimic acid on lipid accumulation by Oil Red O staining in HepG2 and 3T3-L1 cells. (**a**) Effect of shikimic acid on lipid accumulation in HepG2 cells by Oil red staining. Scale bar = 200 μm. (**b**) Effect of shikimic acid on lipid accumulation in 3T3-L1 cells. Shikimic acid was treated for 24 h in HCCs and 3T3-L1 cells. Scale bar = 100 μm. P: Preadipocyte. All experiments were independently performed at least three times. * *p* < 0.05, ** *p* < 0.01.

**Figure 3 ijms-20-00582-f003:**
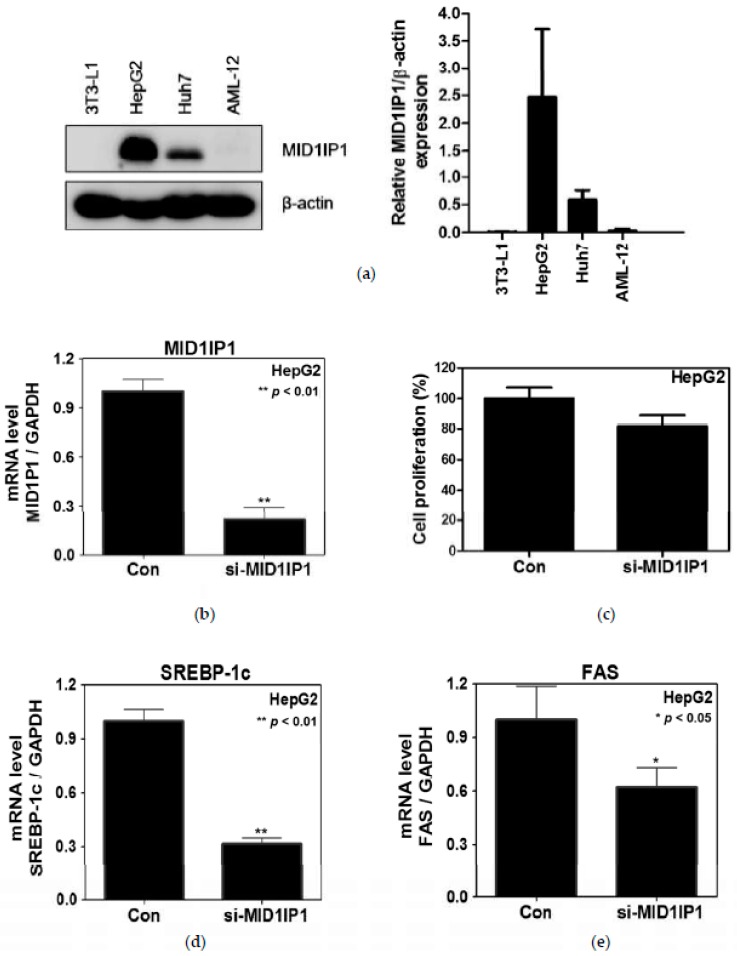
Effect of MID1PI1 depletion on proliferation and lipogenesis-related genes. (**a**) Expression level of MID1IP1 in different cell lines. β-actin was used as loading control. (**b**) Depletion level of MID1IP1 for 48 h in HepG2 cells by qRT-PCR. (**c**) Effect of MID1PI1 depletion on proliferation in HepG2 cells by MTT assay. (**d**,**e**) Effect of MID1PI1 depletion on the mRNA level of SREBP-1c and FAS in HepG2 cells by RT-qPCR analysis. All experiments were independently performed at least three times.

**Figure 4 ijms-20-00582-f004:**
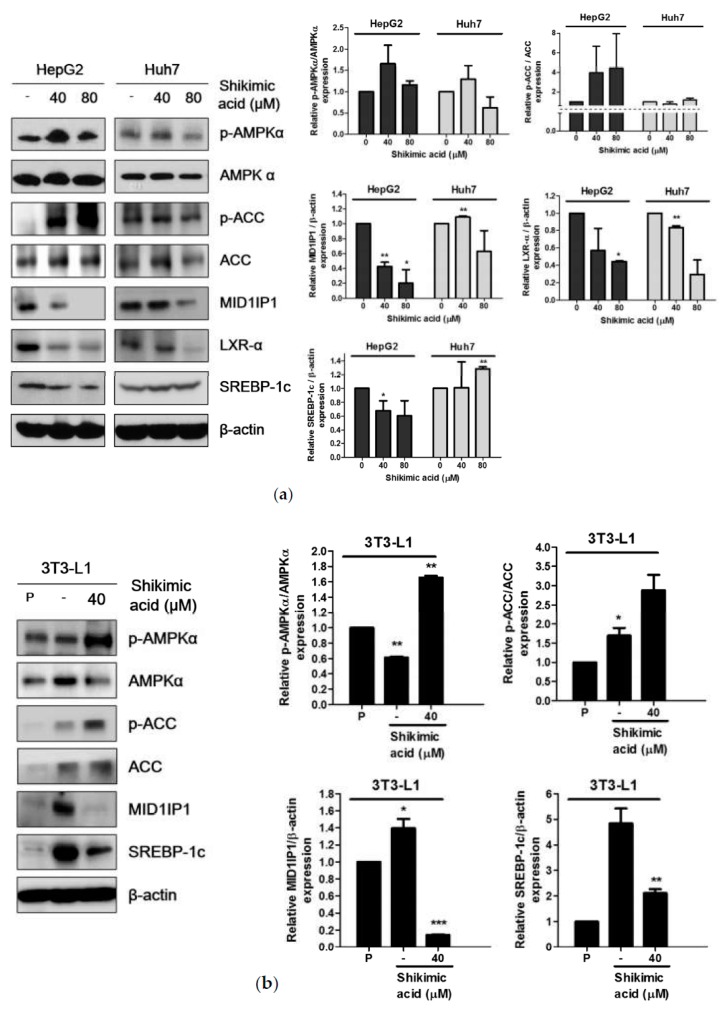
Effect of shikimic acid on lipid metabolism related molecules in HCC and 3T3-L1 cells. Lipogenesis-related proteins were evaluated by Western blotting after treatment of shikimic acid for 24 h in HCCs (**a**) and 3T3-L1 preadipocytes and adipocytes (**b**). P: Preadipocyte. All experiments were independently performed at least three times. * *p* < 0.05, ** *p* < 0.01, *** *p* < 0.001.

**Figure 5 ijms-20-00582-f005:**
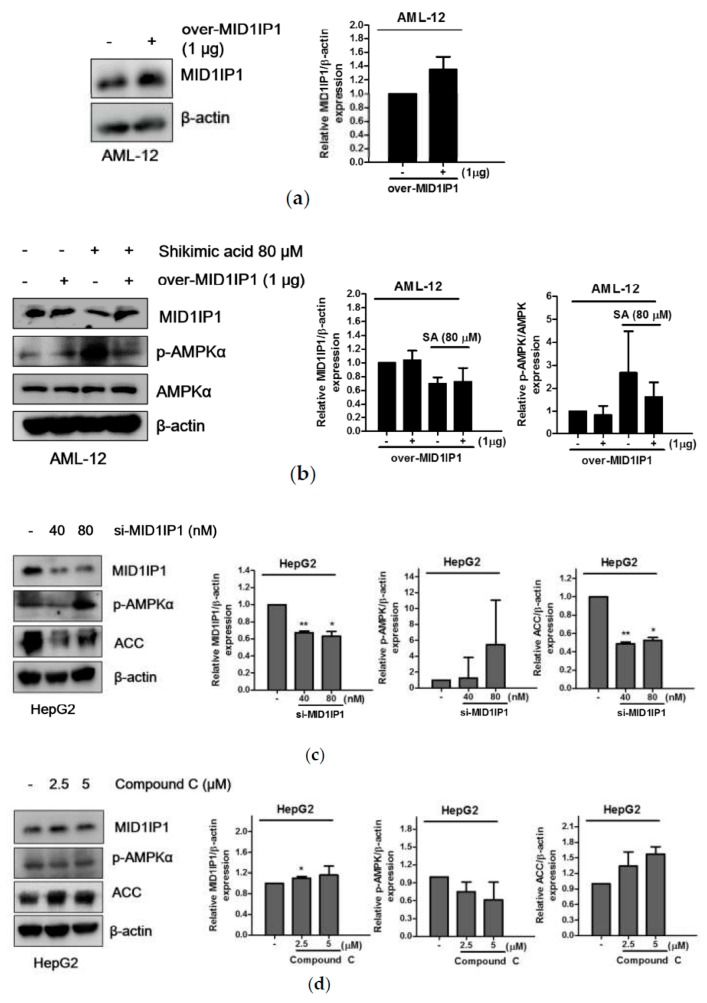
Close relationship between MID1IP1 and AMPK in HepG2 cells. (**a**) Overexpression level of MID1IP1 in AML-12 cells. (**b**) MID1IP1 overexpression for 48 h disturbed phosphorylation of AMPK in AML-12 cells. (**c**) Depletion of MID1IP1 (0, 40, 80 nM) activated phosphorylation of AMPK in HepG2 cells. (**d**) AMPK inhibitor compound c did not affect phosphorylation of AMPK in HepG2 cells. All experiments were independently performed at least three times. * *p* < 0.05, ** *p* < 0.01.

**Figure 6 ijms-20-00582-f006:**
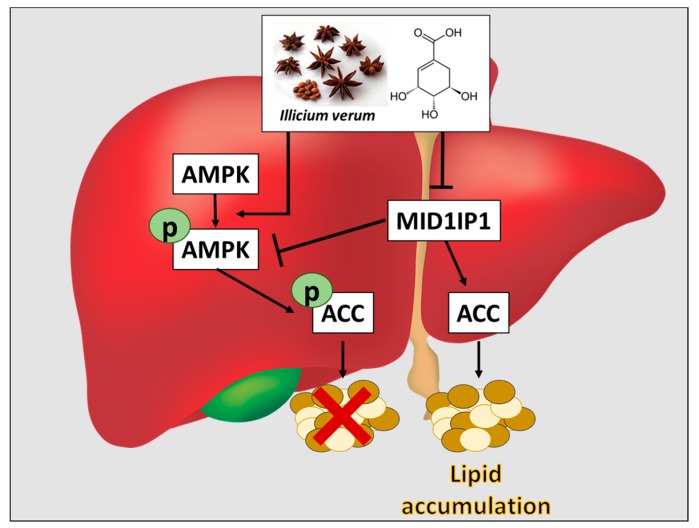
Schematic diagram of hypolipogenic mechanism of shikimic acid via inhibition of MID1IP1 and phosphorylation of AMPK/ACC in hepatocellular carcinomas (HCCs). Black arrow for activation and T bar for inhibition.

**Table 1 ijms-20-00582-t001:** Primers used for quantitative real-time PCR (qPCR) in this study.

	Sense	Antisense
MID1IP1	5′GGC GAC ACC TTT CCT GGA CT3′	5′GAT GGC TGA GGG TGC TCT GT3′
SREBP-1c	5′CCA TGG ATG CAC TTT CGA A3′	5′CCA GCA TAG GGT GGG TCA A3′
FAS	5′GCT GCT CCA CGA ACT CAA ACA CCG3′	5′CGG TAC GCG ACG GCT GCC TG3′

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
