# Peer review of "Hypolipogenic Effect of Shikimic Acid Via Inhibition of MID1IP1 and Phosphorylation of AMPK/ACC"

_ijms, 2019, doi:10.3390/ijms20030582_

Reviewer 1 Report

Very limited research reports the bioactivity of shikimic acid on metabolic diseases. This manuscript reveals that shikimic acid could inhibit lipogenesis via activation of AMPK and MID1IP1 in hepatocytes and adipocytes. The authors provide novel information for the application of shikimic acid on fatty liver disease. However, there are some points need to be clarified or improved.

1. Line 30-32, the sentence is confused. How could induce hepatic steatosis by reduction of fatty acid synthesis since de novo fatty acid synthesis in liver has been mentioned as a key contributor to NAFLD [Postic and Girard, 2008; J. Clin. Invest. 118:829-838].

2. Figure 3, Figure 4, and Figure 5, how many repeats for the experiment was performed? The figure legend should indicate the repeat number of each experiment. The band of western blotting should be quantified and the statistic result should be demonstrated.       

3. Figure 5b, why not to use 40 microM of shikimic acid in the experiment? 80 micro M of SA seems no influence on AMPK phosphorylation.   

4. Line 152, there is no figure 6. 

Author Response

# Reviewer 1

Very limited research reports the bioactivity of shikimic acid on metabolic diseases. This manuscript reveals that shikimic acid could inhibit lipogenesis via activation of AMPK and MID1IP1 in hepatocytes and adipocytes. The authors provide novel information for the application of shikimic acid on fatty liver disease. However, there are some points need to be clarified or improved.

1. Line 30-32, the sentence is confused. How could induce hepatic steatosis by reduction of fatty acid synthesis since de novo fatty acid synthesis in liver has been mentioned as a key contributor to NAFLD [Postic and Girard, 2008; J. Clin. Invest. 118:829-838].

(Response) Corrected as “by promoting de novo fatty acid synthesis,” citing above reference.

2. Figure 3, Figure 4, and Figure 5, how many repeats for the experiment was performed? The figure legend should indicate the repeat number of each experiment. The band of western blotting should be quantified and the statistic result should be demonstrated.      

(Response) Thanks. All experiments were independently performed at least three times. Also, it was quantified based on your comments.

3. Figure 5b, why not to use 40 microM of shikimic acid in the experiment? 80 micro M of SA seems no influence on AMPK phosphorylation.   

(Response) We performed Western blotting with 40 and 80 μM of shikimic acid in AML-12 cells and new blot was added in Figure 5b.

4. Line 152, there is no figure 6.

(Response) Thanks. Graphical Abstract was added in Fig. 6.

Reviewer 2 Report

This manuscript indicates the hypolipogenic effect of shikimic acid via inhibition of MID1IP1 and phosphorylation of AMPK /ACC. The experimental design is logic. However, this manuscript is lacking in several key aspects, numbered below, but with substantial revision could be made suitable for publication.
1. I think Figures 2, 4 and 5 required quantitative analysis and Figure 2 require negative control for HepG2 and Huh7 cells.

2. Authors should draw the mechanism scheme to describe how shikimic acid to reduce lipogenesis through MID1IP1 inhibition and phosphorylation of AMPK /ACC. 3. Authors used several cells such as HepG2, Huh7 and 3T3-L1 adipocyte cells in oil-red staining experiments, HepG2 cells for qRT-PCR and also used HepG2 and AML-12 cells in western blot experiments. I think cell line should be consistent.

4. Authors should describe how long will cause lipid accumulation in HepG2, Huh7 cells and 3T3-L1 cells.

Author Response

# Reviewer 2

This manuscript indicates the hypolipogenic effect of shikimic acid via inhibition of MID1IP1 and phosphorylation of AMPK /ACC. The experimental design is logic. However, this manuscript is lacking in several key aspects, numbered below, but with substantial revision could be made suitable for publication.
1. I think Figures 2, 4 and 5 required quantitative analysis and Figure 2 require negative control for HepG2 and Huh7 cells.

(Response) Thanks. Quantified based on your comments. We included untreated control in Figures 2, 4 and 5.

2. Authors should draw the mechanism scheme to describe how shikimic acid to reduce lipogenesis through MID1IP1 inhibition and phosphorylation of AMPK /ACC.

(Response) Graphical Abstract was added in Figure 6.

3. Authors used several cells such as HepG2, Huh7 and 3T3-L1 adipocyte cells in oil-red staining experiments, HepG2 cells for qRT-PCR and also used HepG2 and AML-12 cells in western blot experiments. I think cell line should be consistent.

(Response) Oil red staining and Western blotting revealed that Huh7cells did not show lipid staining and MID1PI1 expression rather than lipogenic HepG2, cells. Thereafter, we used Hep G2 cells for further study. However, we thought that it is not reasonable to transfect MID1IP1 overexpression plasmid into HepG2 cells, since MID1IP1 is endogenously overexpressed in Hep G2 cells. Thus, used Hep G2 cells for transfection with MID1IP1 siRNA plasmid(Figure 5c), while we used normal hepatocyte AML-12 cells for transfection with MID1IP1 overexpression plasmid (Figure 5b), since MID1IP1 is not endogenously expressed in AML-12 cells.

4. Authors should describe how long will cause lipid accumulation in HepG2, Huh7 cells and 3T3-L1 cells.

(Response) HepG2 and Huh7 cells are lipogenic despite different intensity, while 3T3 L1 cells are lipogenic only after induction of differentiation for 4-5 days.

Reviewer 3 Report

The manuscript is well written, and the results are presented adequately. I recommend this manuscript for publication. Some minor suggestions/comments need to be addressed. Please see the comments in the text of the manuscript as sticky notes.

Author Response

#Reviewer 3

The manuscript is well written, and the results are presented adequately. I recommend this manuscript for publication. Some minor suggestions/comments need to be addressed. Please see the comments in the text of the manuscript as sticky notes.

(Response) Many thanks. We corrected according to your suggestions as shown in sticky notes of pdf review file.

why compound was not used in AML12 cells. HepG2 cells are human cancer cells whereas AML12 cells are normal mouse cells. Normal and cancer cells may expres proteins differently???

(Response) As you know, AML12 cells are normal hepatocytes with low expression of MID1IP1, while HepG2 cells are human lipogenic hepatocarcinoma cells with high expression of MID1IP1 and low expression of p-AMPK.   For efficient experiment, we used HeP G2 cells to examine the effect of AMPK inhibitor on MID1IP1, p-AMPK and ACC.

Please explain why HepG2 cells in fig 5c and 5d show difference in p-AMPK expression, when both are negative control lanes.

(Response) Thanks. We replaced it by new blots, since exposure time can change the expression   level. 

Reviewer 4 Report

This research describes Hypolipogenic effect of shikimic acid via inhibition of MID1IP1 and phosphorylation of AMPK /ACC.

The paper is interesting but, in my opinion, few points should be improved before its publication.

Authors should motivate the choice of the use two different type of hepatocellular carcinoma cell lines (HepG2 and Huh-7)? moreovere, why did you use normal hepatocytes (AML-12) only for the RNA interference and you didn’t use them for all the experiment?  this aspect is crucial

In the abstract could be indicated that shikimic acid reduce lipid accumulation also in Huh-7, as described in paragraph 2.2.

In material and methods, the origin of shikimic acid is not indicated and it must be.

Results of each experiment should be expressed in a clearer and more ordered manner (for example in the title of paragraph 2.5 are indicated 3T3-L1 cells but the discussion is about HepG2 and AML-12)

Line 81: I think it is figure 3b and not figure 2b

Line 152: What and where is the figure 6?

Author Response

# Reviewer 4

This research describes Hypolipogenic effect of shikimic acid via inhibition of MID1IP1 and phosphorylation of AMPK /ACC.

The paper is interesting but, in my opinion, few points should be improved before its publication.

Authors should motivate the choice of the use two different type of hepatocellular carcinoma cell lines (HepG2 and Huh-7)? moreovere, why did you use normal hepatocytes (AML-12) only for the RNA interference and you didn’t use them for all the experiment?  this aspect is crucial

(Response) Thank you for your critical comments. As mentioned in this MS, the focus of this study is the role of MID1IP1 in lipogenic HepG2 cell lines.  For this aim, we first showed HepG2 cells are more lipogenic more than Huh7 cell lines by Oil Red staining. However, we though that it is not reasonable to transfect MID1IP1 overexpression plasmid into HepG2 cells, since MID1IP1 is endogenously over expressed in these cells. Accordingly, we used showed AML-12 cells for transfection with MID1IP1 overexpression plasmid (Figure 5b), and Hep G2 cells for transfection with MID1IP1 siRNA plasmid(Figure 5c). 

In the abstract could be indicated that shikimic acid reduce lipid accumulation also in Huh-7, as described in paragraph 2.2.

(Response) Thanks. Added.

In material and methods, the origin of shikimic acid is not indicated and it must be.

(Response) Shikimic acid was purchased from Sigma.

Results of each experiment should be expressed in a clearer and more ordered manner (for example in the title of paragraph 2.5 are indicated 3T3-L1 cells but the discussion is about HepG2 and AML-12)

(Response) Thanks. We modified description in order.

Line 81: I think it is figure 3b and not figure 2b

(Response) Thanks. corrected.

Line 152: What and where is the figure 6?

(Response) Thanks. . Graphical Abstract was added in Fig. 6.

Round  2

Reviewer 1 Report

none

Reviewer 2 Report

I don't have further questions for the modified manuscript.